# Effects of Nano-CeO_2_ on Microstructure and Properties of WC/FeCoNiCrMo0.2 Composite High Entropy Alloy Coatings by Laser Cladding

**DOI:** 10.3390/nano13061104

**Published:** 2023-03-19

**Authors:** Xiangyu Ren, Wenlei Sun, Zefeng Sheng, Minying Liu, Hujing Hui, Yi Xiao

**Affiliations:** College of Mechanical Engineering, Xinjiang University, Wulumuqi 830047, China

**Keywords:** laser cladding, high entropy alloy, microhardness, wear resistance, corrosion resistance

## Abstract

FeCoNiCrMo0.2 high entropy alloy has many excellent properties, such as high strength, high wear resistance, high corrosion resistance, and high ductility. To further improve the properties of this coating, FeCoNiCrMo high entropy alloy (HEA) coatings, and two composite coatings, FeCoNiCrMo0.2 + WC and FeCoNiCrMo0.2 + WC + CeO_2_, were prepared on the surface of 316L stainless steel by laser cladding technology. After adding WC ceramic powder and CeO_2_ rare earth control, the microstructure, hardness, wear resistance, and corrosion resistance of the three coatings were carefully studied. The results show that WC powder significantly improved the hardness of the HEA coating and reduced the friction factor. The FeCoNiCrMo0.2 + 32%WC coating showed excellent mechanical properties, but the distribution of hard phase particles in the coating microstructure was uneven, resulting in unstable distribution of hardness and wear resistance in each region of the coating. After adding 2% nano-CeO_2_ rare earth oxide, although the hardness and friction factor decreased slightly compared with the FeCoNiCrMo0.2 + 32%WC coating, the coating grain structure was finer, which reduced the porosity and crack sensitivity of the coating, and the phase composition of the coating did not change; there was a uniform hardness distribution, a more stable friction coefficient, and the flattest wear morphology. In addition, under the same corrosive environment, the value of polarization impedance of the FeCoNiCrMo0.2 + 32%WC + 2%CeO_2_ coating was greater, the corrosion rate was relatively low, and the corrosion resistance was better. Therefore, based on various indexes, the FeCoNiCrMo0.2 + 32%WC + 2%CeO_2_ coating has the best comprehensive performance and can extend the service life of 316L workpieces.

## 1. Introduction

High entropy alloys consist of alloying elements with five or more equal molar fractions, each of which has a molar fraction of about 5% to 35%. As a kind of comprehensive material, high entropy alloy has better physical and chemical properties than many common metal compounds [1]. Therefore, it has attracted wide attention in material science and engineering and is the focus field of scholars at home and abroad [2,3]. In recent years, many scholars at home and abroad have used high entropy alloys for surface modification, mainly by laser melting, laser deposition, and plasma melting. The lower preparation cost and excellent performance of high entropy alloys not only improve their service life and significantly reduce the failure loss but also reduce their surface repair cost, and high entropy alloys have many advantages that traditional alloys can hardly match [4]. However, there are difficulties in the synergistic enhancement of strength, toughness, and corrosion resistance of high entropy alloys, and no satisfactory solution has been found so far [5].

WC is a kind of ceramic particle, its hardness is similar to diamond, and it has good electrical and thermal conductivity. WC has stable chemical properties and is often used in the production of cemented carbide materials [6,7]. With the increasing maturity of material surface modification technology, WC is often used in the additive manufacturing field as an additive in powder form to improve the hardness, wear resistance, and corrosion resistance of substrates [8]. Zhang et al. [9] prepared a composite powder coating of TC4 and WC on the surface of TC4 titanium alloy. The results showed that adding 5%, 10%, and 15% WC content could help improve the microhardness and wear resistance of the coating, and the coating forming quality was good without cracks and other obvious defects. Chong et al. [10] added 5% WC into AlCoCrFeNi high entropy alloy powder, and the research showed that WC enhanced the hardness of the AlCoCrFeNi coating, reduced the surface roughness, and improved the wear and corrosion resistance. Hu et al. used high-speed laser cladding to prepare a Ni/WC composite powder coating. The research showed that WC improves the microhardness and wear resistance of the coating and can refine the grain structure, and the decomposition of C and W elements precipitated from the coating played a solid solution strengthening role. Ma et al. [4] studied the microstructure, microhardness, and wear resistance of a 60% WC-reinforced FeCoNiCr composite coating prepared by laser cladding. The results showed that the composite coating had good forming quality and better microhardness and wear resistance.

In addition, due to the fast heating and cooling characteristics of laser cladding technology, it is easy to produce a large temperature gradient between the substrate and the cladding layer, resulting in the occurrence of thermal stress coupled with the existence of a large difference in material properties between the metal substrate and the high entropy alloy coating, which leads to the occurrence of cracks, porosity, and other defects in the coating, which seriously limits the improvement of the coating to the performance of the substrate. Common methods of inhibiting cladding cracking include composite powders, preheating of the substrate, and the application of external field aids such as ultrasound [11,12]. In recent years, a number of experts has carried out a large number of studies on the strengthening of material properties. Their studies found that the powder mixed with rare earth elements can refine the grain and improve the grain boundary state, thus improving the performance of the cladding layer. At present, the most common rare earth powder additives are CeO_2_, Y_2_O_3_, La_2_O_3_, and so on [7,13,14,15,16,17,18]. Zhang et al. [19] incorporated nano-CeO_2_ particles as additives into Ni625 alloy powder and compared the differences in microstructure and electrochemical and tribological properties of Ni625 and Ni625 + 0.2% CeO_2_ fused cladding layers. The results showed that nano-CeO_2_ could inhibit the liquefaction crack of Ni625 coating, refine the grain, and improve the corrosion resistance and wear resistance. Liu et al. prepared a composite coating of Ti-based and TC4 + Ni60 powders on Ti811 alloy plates, and then after the addition of 2% rare earth CeO_2_, the results showed that the addition of CeO_2_ greatly improved the surface quality of the coating and promoted the precipitation of TiC particles. Conversely, excessive CeO_2_ can hinder the refinement of the microstructure and the improvement of microhardness, thus slightly reducing the wear resistance of the coating [20]. Gao et al. [14] studied the effects of CeO_2_ addition on crack sensitivity, microstructure, phase composition, solute segregation, and microhardness of the Ni60 cladding layer. Research shows that the addition of 4.0% CeO_2_ can effectively suppress cracks and porosity in the Ni60 clad layer, promote grain refinement, and improve tissue uniformity, but too much CeO_2_ will cause lattice distortion in each phase of the coating. Zhang et al. [21] added CeO_2_ powder to Ti6Al4V + NiCr-Cr_3_C_2_ composite powder to study the effect of CeO_2_ on the forming quality and microstructure of a TiC_x_-reinforced CrTi_4_-based composite coating by laser cladding technology. The results showed that the content of TiC_x_ dendrites is closely related to the CeO_2_ content. The addition of CeO_2_ has a significant effect on the cracking, porosity, and dilution ratio of the laser cladding Ti6Al4V/NiCr-Cr_3_C_2_ composite coating, but it has no effect on the phase composition.

As is well known, in today’s industrial industry, 316L stainless steel is widely used in the automotive, aircraft, aviation, and other fields due to its superior overall performance. However, these various parts, with 316L as the material, are continuously subjected to harsh working environments such as frictional wear, air, or seawater in actual engineering, such as the connecting flanges in many wind turbines. After a long period of work, these parts are subject to forms of failure such as wear, spalling, and pitting, which all occur on the surfaces of the workpieces. Thus, the purpose of this study is mainly to improve the physical and chemical properties of 316L stainless steel by surface modification of this material using laser melting, thus extending the service life of 316L stainless steel and promoting the industrialization of sustainability.

Based on these, two kinds of composite cladding layers, namely, FeCoNiCrMo0.2 + WC and FeCoNiCrMo0.2 + WC + CeO_2_, were prepared on the surface of 316L stainless steel by laser cladding technology using mechanical self-blending powder as raw materials [22]. The effects of WC and CeO_2_ on the phase structure, microstructure, and microhardness of the cladding layer of high entropy alloy were studied to provide theoretical reference and experimental examples for subsequent research on the preparation process of improving the properties of high entropy alloy.

## 2. Materials and Methods

The base material used was 316L stainless steel [23,24,25], size 150 × 50 × 8 mm. The surface to be fused was polished with 150# and 600# sandpaper to remove oxides and rust traces, and then it was cleaned with anhydrous ethanol to remove surface oil and dirt [26]. Using FeCoNiCrMo0.2 high entropy alloy powder [1,4] with a purity higher than 99.98% as raw material, the composition is shown in Table 1, adding 8%, 16%, 24%, 32%, and 40% of WC spherical powder, respectively [6,9]; powder particle sizes ranged from 15 to 53 μm [27]. Furthermore, the powder was ball milled for 3 h using a KQM-Z/B type planetary ball mill to mix the powder fully and uniformly. The grinding material was agate ball, the ball/material ratio was 3:1, and the rotation speed was 350 r/min. After mixing, the powder was placed in a drying oven for 30 min to remove the moisture from the air, and then it was cooled to room temperature and then placed in a sealed bag for use [26].

In this experiment, a high-speed GS-CW-6000G-91-202 fiber laser made by Shanxi Guosheng Laser Technology Company was used to process the single and multi-channel samples of high entropy alloy coating [5]. Coaxial powder feeding processing was adopted, and argon gas was used as the protective gas. The process parameters were optimized by the orthogonal test, and the specific values are shown in Table 2 [28,29,30,31]. The laser cladding equipment [13] and system components are shown in Figure 1.

To begin with, a single-channel cladding experiment of FeCoNiCrMo0.2 + WC composite powder with various proportions was conducted, and it was found that the additive amount of WC with the optimal coating morphology was 32%. Moreover, the same steps were followed to add different contents [32] of CeO_2_ rare earth oxide to the composite powder, respectively [1]. The chemical composition of the composite coating is shown in Table 3. The particle size of CeO_2_ powder ranged from 40 to 80 nm [33].

After comprehensive consideration of cladding morphology, dilution rate, microhardness, friction coefficient, and other indicators, it was finally determined that FeCoNiCrMo0.2 + 32%WC + 2%CeO_2_ was the best composite cladding layer. The SEM morphology of the powders is shown in Figure 2 [13,28,34].

Then the three powders, namely, FeCoNiCrMo0.2, FeCoNiCrMo0.2 + 32%WC, and FeCoNiCrMo0.2 + 32%WC + 2%CeO_2_, were used as cladding powder to conduct single-channel and multi-channel tests. The multi-channel and single-channel laser cladding samples adopted the same process parameters. The microstructure, phase composition, microhardness, and wear and corrosion resistance of the coatings were compared.

After the single-channel cladding sample was completed, the sample was divided into 10 × 10 × 8 mm by wire cutting. Then the cross-section of the specimen was finely polished by an MP-2B grinding and polishing machine with sandpaper grit sizes of 800#, 1000#, 1200#, 1500#, and 2000# in order. This polishing machine was used to polish the cross-section of the sample. The composition of the polishing agent was polycrystalline diamond powder, and the particle size of the polishing agent was W0.5 μm. After polishing, industrial alcohol was used to scrub off the polish and other impurities on the cross-section, and then aqua regia (volume ratio HCl:HNO_3_ = 3:1) was used to carry out metallographic corrosion on the cross-section for 30 s, and the etching was done by dripping with a rubber-tipped dropper. The macroscopic morphology and dilution rate of the cross sections of coatings were analyzed by an industrial microscope. The phase analysis of the coating was carried out by a Brock D8 Advance (Germany) X-ray diffractometer (XRD). A JSM-7610F plus scanning electron microscope (SEM) and energy dispersive spectrometer (EDS) were used to observe the morphology and distribution of elements. The microhardness was measured by a Huayin HV-1000A digital microhardness tester. The applied force was 200 N, and the residence time was 15 s. An MFT-5000 friction and wear tester was used to test the friction coefficient, and Si_3_N_4_ ceramic balls were used as friction subsets for dry friction experiments under a load of 10 N, with a loading time of 30 min. The corrosion test of the coating was carried out by using a CHI660D electrochemical workstation, the reference electrode was saturated glycury electrode, the counter electrode was platinum electrode, and the specimen was the working electrode. The corrosion medium was 3.5% NaCl solution with pH = 7, the initial potential was −1 V, the termination potential was 0 V, and the scanning speed was 5 mv/s.

## 3. Results

### 3.1. Morphology Analysis of Cladding Layers

Figure 3 shows the macroscopic morphology and cross-sectional morphology of the composite coatings prepared with different powder compositions observed under an industrial microscope.

It can be seen from the figure that the forming quality of cladding layers prepared with different powder materials was different. The surface of the FeCoNiCrMo0.2 (HEA) cladding layer was flat, and no obvious cracks were found. There was no good metallurgical bonding between the coating and the substrate. After the addition of 32% WC, the dilution rate of the coating increased, and the metallurgical bonding between the coating and the substrate was relatively good. The macro-morphology of the FeCoNiCrMo0.2 + 32%WC (HEA/WC) coating was better than that of the FeCoNiCrMo0.2 coating. Nevertheless, when the WC content increased to 40%, a small number of cracks and obvious pores appeared in the coating, which is because a large number of WC particles entered the molten pool and were burned and decomposed to form W_2_C and C, and the C atom reacted with the O atom in the air to precipitate CO and CO_2_. The chemical formula [15] is shown in Equations (1) and (2):(1)$$2\mathrm{WC}\to W_{2}C+C$$
(2)$${3C+2O}_{2}\to {2\mathrm{CO}+\mathrm{CO}}_{2}$$

Of course, part of the reason is that a small amount of metal vapor gas did not float up and out during the solidification of the melt pool and remained in the coating to form pores.

After adding 2% CeO_2_ rare earth, the FeCoNiCrMo0.2 + 32%WC + 2%CeO_2_ (HEA/WC + CeO_2_) coating had no obvious bubbles, cracks, and other defects. This was due to the strong affinity of rare earth elements with O elements, which tended to combine preferentially with O atoms in the melt pool or air to form rare earth oxides, thus reducing the number of pores in the coating. In addition, the thickness and molten pool depth of the coating were shallow, and the dilution rate was decreased, so that the metallurgical bonding was good. The dilution rate reflected the combination of coating and substrate. Equation (3) shows that the dilution rates η of HEA, HEA/WC, and HEA/WC + CeO_2_ coatings were 60.3%, 52%, and 48.3%, respectively. In the formula [1,35,36], η denotes dilution rate, S_2_ denotes the molten area of the 316L substrate, S_1_ denotes the area of the coating [1], H_1_ denotes the coating thickness, and H_2_ denotes the molten pool depth.
(3)$$\eta =\frac{S_{2}}{S_{1}+S_{2}}\times 100\%=\frac{H_{2}}{H_{1}+H_{2}}\times 100\%$$

However, excessive rare earth will increase the absorption of the beam energy by the molten pool, resulting in a rise in the temperature of the molten pool and a greater probability of the matrix melting, resulting in an excessive dilution rate of the coating in the process of cladding. In addition, the coating will appear to be an obvious oxidation layer, which is caused by the reaction of O element with Fe and other metal elements.

### 3.2. Coating Phase Analysis

The X-ray diffractometer (XRD) of the German Brock D8 Advance was used to scan the coating for phase analysis. The voltage was 40 kv, the current was 40 mA, the step size was 0.02, and the step speed was 0.2 s/step. Then Jade software version 6.5 was used to analyze the XRD results. Figure 4 shows the XRD patterns of composite coatings prepared by three different powders [13].

Based on the XRD pattern, the evaluation results show that the FeCoNiCrMo0.2 coating and the original powder phase lattice structure were the same, namely, α-Fe, Cr, and Mo elements of the body-centered cubic (BCC) lattice structure [1], Ni and Co elements of the face-centered cubic (FCC) lattice structure [5], and did not generate a new phase composition. After the addition of WC, new diffraction peaks appeared, which were mainly due to WC and its decomposition to generate trace intermetallic compounds (laves phase) and carbides, such as W_2_C, MoC, Mo_2_C [15], and other dense rows of hexagonal closest packed (HCP) lattice structure and complex dotted structure compounds, such as Cr_7_C_3_, Fe_3_W_3_C, and Ni_2_W_4_C [13,37]. Some scholars have shown that after WC particles enter the molten pool, they will burn and decompose to form W_2_C and C. At the same time, WC reacts with solid solution metal atoms (Cr, Fe, Ni, Mo, etc.) under high temperature to form carbide M_2_C, M_7_C_3_, and M_23_C_6_ with low melting points. The chemical formula [4] is shown in Equation (1) and Equation (4).
(4)$$\mathrm{WC}+M\to M_{2}{C+M}_{7}C_{3}{+M}_{23}C_{6}+\delta$$

In the formulas, δ is other trace metal compounds, and M is any combination of Cr, Fe, Ni, Mo, Co, and other elements. After the addition of CeO_2_ rare earth elements, the number of diffraction peaks decreases and is the same as that of the FeCoNiCrMo0.2 coating. This indicates that the addition of CeO_2_ not only does not generate new cladding layer phases, but it also inhibits the generation of dense hexagonal, complex grain structure carbides generated by the chemical reaction between WC and solid solution metals, and it has a significant effect on the generation of each intermetallic compound phase. The diffraction peak positions of each phase in the coating also did not change significantly after the addition of rare earths, which means that the rare earth oxides did not participate in the internal reactions of the phases or crystals. The addition of CeO_2_ restores the BCC phase diffraction peak in the layer, and the peak is close to that of the FeCoNiCrMo0.2 coating. The addition of 2% CeO_2_ decreased the diffraction peaks of complex dotted compounds, and according to the Debye–Scherrer formula [14] in Equation (5), the addition of CeO_2_ reduced or refined the grain size.
(5)$$D=\frac{K\lambda }{\beta \cos\theta }$$

In the Equation (5), *D* is the grain size, K is the Scherrer constant, *λ* is the wavelength of the X-ray, *β* is the half-height width of the diffraction peak, and *θ* is the Bragg diffraction angle.

Observing the XRD diffraction spectrum with the addition of 2% CeO_2_, no diffraction peaks related to CeO_2_ and other Ce elements were found, indicating that the content of CeO_2_ was low and not enough to generate new physical phase diffraction peaks in the XRD technique, so the Ce elements existed in the microstructure in the form of a solid solution. In the XRD pattern with 2% CeO_2_, the BCC phase decreased at 2θ = 50°, while the BCC and FCC phases increased at other positions.

### 3.3. Microstructure of the Coating

A JSM-7610F plus a scanning electron microscope (SEM) and an energy dispersive spectrometer (EDS) were used to observe the morphology and distribution of elements. The SEM was used to scan the microscopic morphology of different regions of the three coatings [13,38]. Figure 5 shows the microstructure of the three coatings at the top, middle, and bottom, respectively.

As can be seen from Figure 5a–c, the top microstructure of the HEA coating was mainly composed of flocculent grains; the microstructure in the middle of the coating was reticular, and the grain grew directionally; the bottom of the coating was mainly composed of reticular grains, but dendritic grains also appeared, which were the result of the reaction among five principal elements in the coating during the cladding process and the formation of trace metal compounds after cooling [5]. There were no defects in the HEA coating, such as pores and cracks [11].

As can be seen from Figure 5d–f, with the addition of WC content, the grain morphology also changed due to the change of phase, and secondary crystal axis began to be formed. This is because a host of metal compounds and carbides, such as W_2_C, Cr_7_C_3_, and Fe_3_W_3_C, were produced after the addition of WC, which changed the original microstructure of the HEA coating. According to Figure 5d,e, the top microstructure of the HEA + 30%WC coating was a fine and multi-directional dendrite structure [1], and the dendrite region on the top was not clearly demarcated. The HEA/WC coating had a dendritic structure with directional growth in the middle. Compared with Figure 5b,e, HEA/WC had a more ordered and finer eutectic structure. As can be seen in Figure 5f, the microstructure at the bottom of the coating transformed into cellular crystals, and carbide particles appear. This indicates that the addition of WC particles and their decomposition formed intermetallic compounds with carbides, which provided more nucleation masses, led to non-uniform nucleation during solidification, hindered the growth of dendrites and obtained fine grains, and refined the grains of the HEA coating, and there were almost no unmelted and decomposed WC particles in the coating. Conversely, it also caused a difference in the coefficient of thermal expansion in the coating, which reduced the fluidity of the molten pool and produced a small amount of porosity. During solidification, the microscopic grain structure is related not only to the temperature gradient (G) between the solid–liquid interface but also to the rapid solidification rate (S) during the melting process, where the value of S gradually increases from the bottom to the top of the coating. The larger the G/S value is, the coarser the grain structure is, while the smaller the G/S value, the finer the grain structure [1,39].

As seen in Figure 5g–i, the addition of CeO_2_ further refined the grain structure of the coating, and the grain structure at the top and bottom turned into cellular crystals with multi-directional grain growth. This is because, compared with Fe, Co, and other elemental elements in the HEA/WC coating, the Ce atom has high chemical activity and a relatively large atomic radius, which tends to cause lattice distortion and increases the free energy of the system. In addition, the Ce atom in a molten pool does not exist in the form of elemental elements, but will it form finer CeO_x_ particles as nucleating points and enrich at grain boundaries. The Ce gathered at grain boundaries inhibits the further growth of grains and has a pinning effect on the movement between grain and the phase interface, thus refining the microstructure. This is evidenced by the significant change of the porous particle size, from large particles from the top of the HEA/WC coating in Figure 5d to the fine particles in Figure 5g. As can be seen from Figure 5h, the grain structure in the middle transformed from dendritic to cellular, and the grain boundary was more obvious and the grain distribution more dense [16]. This is because the addition of CeO_2_ can inhibit the formation of many metal compounds and carbides produced by the addition of WC, which affects the microstructure of the coating. As shown in Figure 5i, at the bottom of the coating, planar crystals could be found, with a large temperature difference between the coating and the 316L substrate, forming finer cellular crystals under directional solidification.

### 3.4. Element Distribution

Energy spectroscopy and scanning elemental distribution of two coatings, namely, HEA/WC and HEA/WC + CeO_2_, were performed using an energy dispersive spectrometer (EDS). Figure 6 and Figure 7 show the results of energy spectrum analysis [13] and their elemental distribution for the surface sweep of the top part of the HEA/WC and HEA/WC + CeO_2_ coatings, respectively [18,21,40].

It can be seen from Figure 6a that WC was evenly distributed in the HEA coating, indicating that WC was decomposed under the thermal effect of the laser beam and the liquid molten pool. As can be seen in the figure, a small number of pores were generated in the coating after WC addition, which proves that WC causes a difference in the coefficient of thermal expansion. It can be seen from Figure 6b that the atomic content of element C was 27.04 at.%, while that of element W was only 5.51 at.%; the reason may be that the temperature of the molten pool exceeded the melting point of WC particles, causing some WC particles to dissolve and some W elements to burn out. In addition, the percentage of each element in the coating deviated from the design ratio of the high entropy alloy coating; combined with XRD analysis, this was also because after the addition of WC, a eutectic reaction occurred with metal elements such as Fe in the cladding process, resulting in a large number of liquid phase intermetallic compounds and carbides, such as M_2_C, M_3_C, and M_23_C_6_. When left to cool, a large number of eutectic precipitates were produced, such as a mixture of M_23_C_6_, WC, and W_2_C, resulting in atomic occupancy ratios that were not consistent with the ideal. The chemical formula of the eutectic reaction [10] is shown in Equations (6) and (7).

At about 1150 °C,
(6)$$3\mathrm{Fe}+C\to {\mathrm{Fe}}_{3}C\;(\mathrm{liquid}\;\mathrm{phase})$$

After cooling,
(7)$${\mathrm{Fe}}_{3}C\;(\mathrm{liquid}\;\mathrm{phase})\to \gamma +{\mathrm{Fe}}_{3}C\;(\mathrm{cementate})$$

Moreover, based on Figure 6b and Table 4 [26], it was concluded that among the five principal metal elements of the HEA coating, Fe and Cr elements were more abundant [24], which was due to the small absolute value of the mixing enthalpy of Fe and Cr with four other metal elements, so they did not easily form compounds with other metal elements but tended to segregate among dendrites. Related studies have shown that the more negative the enthalpy of mixing between atoms, the easier it is to form compounds. In other words, the value of the enthalpy of formation can reflect the affinity of elements to some extent. Furthermore, the high content of Cr was also related to the 316L substrate, because the content of Cr in the 316L stainless steel substrate was high, and under the action of the high energy laser beam, part of the Cr element diffused into the coating, resulting in a high content of Cr in the coating.

As can be seen in Figure 7a, the distribution of C and Ce elements in the EDS region was uneven, and the reasons are speculated to be as follows: Firstly, the powder mixing time was not enough in the preparation of the composite powder, resulting in uneven powder distribution. Secondly, the eutectic reaction between the C element and some metal elements in HEA generated a great deal of eutectic crystals. These eutectic crystals contained plenty of the C element, resulting in uneven distribution of the C element. Thirdly, Ce, as a typical surfactant, is easily captured by crystal defects in the cladding process, so a large amount of Ce atoms was adsorbed between the grain boundary and the phase boundary and were not uniformly distributed in the cladding layer.

As shown in Figure 7b, after appropriate addition of the Ce element, a large number of Ce atoms adsorbed between grain boundaries and phase boundaries, enhancing the pegging effect. Owing to the pegging effect in the laser molten pool, the phase boundary migration obstructed the aggregation of adjacent WC particles, thus affecting the microstructure of WC in the solidification process, forming uniformly dispersed fine WC particles in the coating [39] and inhibiting the generation of intermetallic compounds and carbides, so that their metal elements were evenly dispersed in the coating. Therefore, compared to the coating without CeO_2_ addition, the content of metal elements, such as Fe and Ni, was slightly increased. In summary, the appropriate amount of rare earth elements can play a role in refining the grain size and enhancing the mechanical properties of the composites.

### 3.5. Microhardness

A Huayin HV-1000A digital microhardness tester was used to measure the microhardness of the substrate and three kinds of coatings. The applied force was 200 N, and the residence time was 15 s [19]. The obtained hardness distribution curves [1,13,35] are shown in Figure 8.

Figure 8 depicts the microhardness distribution curves for the three different coatings. As can be seen from the figure, the level of microhardness of the coating was clearly different in the coating zone, heat affected zone, and the substrate. In the coating area, the maximum microhardness of the HEA coating was 470.6 HV_0.2_, while the maximum hardness values of the HEA/WC and HEA/WC + CeO_2_ coatings were 562.47 HV_0.2_ and 517.76 HV_0.2_, respectively, an increase of about 100 HV_0.2_ and 50 HV_0.2_, respectively. In the HAZ (heat affected zone), the microhardness in this area decreased significantly, as the 316L substrate was diluted into each coating, but the coating maintained its properties. As a result, the heat affected zone was harder than the 316L substrate. As can be seen from Figure 8, the hardness of the 316L substrate was about 220 HV_0.2_, and the hardness of the HEA/WC coating was about 2.6 times that of 316L and 1.2 times that of the HEA coating. This suggests that WC particles can significantly improve the microhardness of HEA coatings for the purpose of surface strengthening.

The improvement of microhardness is mainly due to solution strengthening and dispersion strengthening. Combined with XRD diffraction pattern, it can be seen that the mixing entropy of the HEA coating was very high, and the coating was the solid solution composition of simple phases BCC and FCC. Moreover, more elements C and W were precipitated from the molten WC particles, which promoted the formation of carbide and the W-rich phase in the cladding layer, producing a solid solution strengthening effect on the coating and improving the microhardness of the coating. However, while the hard M_y_C_x_ and Laves phases significantly strengthened the microhardness of the HEA coating, they also increased the brittleness of the coating and contributed to the generation of cracks within the coating. Hence, although the addition of WC particles significantly improved the hardness of the coating, the hardness curve was unstable, and the internal hardness fluctuated significantly, which was due to the uneven distribution of WC particles in the HEA/WC coating area, resulting in an uneven distribution of the hard phase. The addition of CeO_2_ rare earth elements hindered the formation of the reinforcing phases such as high hardness Cr_23_C_6_, Fe_5_C_2_ and FeNi_3_ in the cladding layer, so the microhardness of the HEA/WC + CeO_2_ coating was lower than that of the HEA/WC coating. Alternatively, the addition of CeO_2_ inhibited the occurrence of the eutectic reaction, reduced the generation of eutectic compound particles, effectively refined the grain organization, and uniformly distributed the microstructure, thus improving the toughness of the molten cladding layer. The microhardness of the HEA/WC + CeO_2_ coating was about 1.1 times that of the HEA coating, and the hardness of the coating was more stable. As the distance from the coating to the substrate decreased, the hardness increased first and then decreased.

### 3.6. Wear Resistance Test

An MFT-5000 friction and wear testing machine was used to test the friction coefficient of the coating. A Si_3_N_4_ ceramic ball was used as the friction pair, the dry friction experiment was carried out under a load of 10 N, and the loading time was 30 min. The results of the wear test are shown in Figure 9.

Figure 9a,b show the friction coefficient curves of three kinds of coatings and substrate with the change of wear time and the wear weight loss of samples [13,21] before and after the test. This shows that there were two phases of initial running-in and stable break-in for all three coatings. Among them, the friction coefficients of HEA and HEA/WC + CeO_2_ coatings tended to level off at the stage when the wear time was about 5 min. However, the HEA/WC coating gradually moved to a stable stage after about 15 min. This is because after the HEA coating was added to WC, a large number of hard phase structures appeared in the coating, such as Cr_7_C_3_ and Fe_3_W_3_C, which improved the hardness and wear resistance of the coating. These hard compounds were not evenly distributed, resulting in a friction coefficient that could not be leveled off in a short time.

The addition of CeO_2_ inhibited the production of these hard phase compounds and refined the structure, so the friction coefficient increased slightly, but the wear property remained stable in the test. Even if in the first half of the test, due to the influence of hard compounds, the wear amount of HEA/WC coating was small and the friction coefficient decreased, with the progress of the friction and wear test, the hard compounds were gradually worn, and the friction coefficient of the coating gradually increased. Finally, the friction coefficient of the coating after leveling off was slightly greater than that of the HEA/WC + CeO_2_ coating. In the friction wear experiment, because the specimen surface was not flat enough, collision friction occurred between the Si_3_N_4_ ceramic ball friction pair and the micro-convex body on the specimen surface, and the contact stress caused the friction coefficient to fluctuate. As the friction wear continued, the protruding part of the coating surface was gradually smoothed, the actual contact area between the friction sub and the coating increased, and the friction factor tended to stabilize and entered the stable break-in phase. The collision friction occurred between the micro-convex bodies on the specimen surface, and the larger contact stress resulted in the fluctuation of the friction coefficient. With the progress of friction and wear, the protruding part was gradually worn down, the actual contact area increased, the friction factor tended to be stable, and it entered the stable break-in stage [2].

The tests showed that under the same load, the average friction coefficients of 316L substrate, HEA, HEA/WC, and HEA/WC + CeO_2_ coating were 0.82, 0.61, 0.29, and 0.43, respectively. The wear amounts of the samples were 3.364 mg, 2.642 mg (26.52% less than that of the substrate), 1.564 mg (53.51% less than that of the substrate), and 2.225 mg (33.86% less than that of the substrate), respectively.

Figure 10 shows the SEM microscopic morphology of the 316L substrate with HEA, HEA/WC, and HEA/WC + CeO_2_ at different multiples after wear of the three coating surfaces, respectively [13,35]. The main wear mechanisms in coatings include adhesive wear, abrasive wear, and oxidation wear. Figure 10a–c show that the wear characteristics of the 316L substrate mainly included a host of spalling, grooving, and a small amount of oxidation. There was a large number of irregular pits and a few grooves attached with debris on the substrate wear marks. This is because of the plastic deformation caused by the adhesion between the substrate and the friction pair; the adhesion point tearing produced a large number of pits and abrasive chips, and the wear resistance was poor. As seen in Figure 10d–f, the HEA coating produced plastic deformation, cracks, and slight grooves on the coating surface after the wear test. The coating wear mechanism was primarily adhesive wear. Compared to the substrate, the HEA coating had superior wear resistance. In Figure 10g–i, it can be seen that in the wear morphology of the HEA/WC coating, only a small amount of spalling and slight grooves occurred due to the high hardness. Owing to the production of hard compounds in the coating, the wear mechanism was transformed into abrasive wear. Compared with the HEA coating, the HEA/WC coating had better wear resistance and a narrower width of wear marks. As seen in Figure 10j–l, the morphology of the HEA/WC + CeO_2_ coating was the best after wear, and the wear width was the narrowest. The wear characteristics of the coating were mostly grooves without spalling or cracking. This shows clearly that CeO_2_ can indeed refine the grain structure, reduce the cracking sensitivity of the coating, improve the wear resistance and toughness of the coating, and extend the service life of the workpiece.

### 3.7. Electrochemical Corrosion Resistance Test

The CHI660D electrochemical workstation was used for corrosion testing of the substrate as well as the three coatings, with the reference electrode being a saturated glycerol electrode, the counter electrode being a platinum electrode, and the specimen being the working electrode. The corrosion medium was a 3.5% NaCl solution with pH = 7, the initial potential was −1 V, the termination potential was 0 V, and the scanning speed was 5 mv/s. The electrochemical corrosion test results obtained are shown in Figure 11 as well as in Table 5 [26].

Figure 11a shows the measured electrochemical polarization curves [1] of the 316L substrate, HEA, HEA/WC, and HEA/WC + CeO_2_ coatings. The polarization curves in Figure 11a represent the pitting resistance of the three metal coatings and the 316L substrate in 3.5% NaCl electrolyte at pH = 7. The four polarization curves in the Figure 11a indicate that the three coatings and the substrate could all form stable passivation intervals in the electrolyte, and the passivation phenomenon could increase the corrosion resistance of the coating. Table 5 shows the self-corrosion potential E_corr_ and the self-corrosion current density I_corr_ obtained by Tafel fitting of the polarization curves. E_corr_ reflects the possibility of corrosion of the coating [37]. The more positive it is, the less probability there is of corrosion occurring. I_corr_ reflects the corrosion rate of materials. The smaller the I_corr_ value is, the slower the corrosion rate will be. According to Table 5, the self-corrosion potential E_corr_ value of the three coatings was positive compared to that of the substrate, and the self-corrosion current density value I_corr_ was one order of magnitude smaller than that of the substrate. The comparison in Table 5 shows that the self-corrosion potential E_corr_ values of all three coatings were positive with respect to the substrate, and the self-corrosion current density values I_corr_ were all one order of magnitude smaller than the substrate. The passivation of the HEA/WC + CeO_2_ coating occurred at a voltage of −0.684 V, and the passivation zone of the polarization curve of this coating was the most obvious and the corrosion current density was the lowest, indicating that the HEA/WC + CeO_2_ coating produced the densest passivation film during the corrosion process and had the best corrosion resistance.

Figure 11b,c show the Nyquist diagram and Bode plot obtained by fitting the electrochemical impedance spectroscopy (EIS) of the four materials. The equivalent circuit is also displayed in Figure 11b, where R_s_ is the resistance of the 3.5% NaCl solution, R_f_ is the resistance of the passive film on the surface of the sample, C_CPE1_ is a constant phase element (CPE) associated with the capacitance between the passive film and 3.5% NaCl, R_ct_ is the transfer resistance between the sample and the 3.5% NaCl solution while the passive film was damaged, and C_CPE2_ is the capacitance at this time. The fitting results of components in the equivalent circuit are listed in Table 6, where n_1_ and n_2_ are, respectively, the dispersion coefficients of CPE1 and CPE2. The closer the n value is to 1, the more equivalent the CPE is to the ideal capacitance. The order of magnitude of χ^2^ was 10^−3^, which indicated that the selected equivalent circuit was reasonable, and the fitting result was reliable. R_p_ is the polarization impedance of the samples in a 3.5% NaCl solution, calculated as shown in Equation (8).
(8)$$R_{P}=R_{S}+R_{f}+R_{ct}$$

Figure 11b shows that adding CeO_2_ can significantly increase the capacitance arc diameter of the coating, indicating that charge transfer is more difficult at the interface between the electrolyte and electrode, and the pitting resistance is superior, which is consistent with the potentiodynamic polarization curve results [10]. Figure 11c shows the relationship among the frequency, impedance modulus, and phase angle of the four samples. As can be seen from the plot, in the 3.5% NaCl solution, the impedance modulus (|Z|) of all four materials in the high-frequency region (10^4^–10^5^ Hz) was about 5 Ω · cm^2^, and in the low-frequency region (10^−2^–10^−1^ Hz), the impedance modulus of the HEA/WC + CeO_2_ coating was 1.25 and 1.48 times higher than that of the HEA/WC coating and the substrate, respectively, which indicates that the HEA/WC + CeO_2_ coating had greater transfer resistance and higher passivation film density, and Clˉ ions could hardly penetrate the passivation film to corrode the sample, so the coating had better corrosion resistance. In the intermediate frequency region (10^−1^–10^4^ Hz), the breakdown resistance of the passive film was related to the phase angle [1]. As shown in Figure 11c, the phase angle in the Bode plot changed from 51° to 60° in the HEA/WC coating after CeO_2_ was added. The phase angle of the HEA/WC + CeO_2_ coating was the largest, indicating that the coating generated the highest density of the passivation films. As shown in Table 6, the polarization impedances (R_p_) of the 316L substrate, HEA, HEA/WC coating, and HEA/WC + CeO_2_ coating were 1.007 × 10^3^ Ω · cm^2^, 1.051 × 10^3^ Ω · cm^2^, 1.12 × 10^3^ Ω · cm^2^, and 1.48 × 10^3^ Ω · cm^2^, respectively [1]. Therefore, after comprehensive evaluation, under the same corrosive environment, the HEA/WC + CeO_2_ coating had a relatively lower corrosion rate, better corrosion resistance, and better protection for the 316L substrate.

## 4. Conclusions

(1) The HEA/WC (FeCoNiCrMo + 30%WC) coating prepared by laser cladding technology has more nucleation particles than HEA (FeCoNiCrMo0.2) coating, and WC is distributed at the grain boundary, which can delay the growth of grain, refine the microstructure of the coating and improve the morphology of the coating. As the metallic elements in the coating tend to react with WC, it makes a host of hard compounds in the coating and produces more physical phase structure. The addition of 2% nano-CeO_2_ rare earth oxide reduces the diffraction peaks of the HEA/WC coating and suppresses the generation of compounds to a certain extent without generating new phases, and the grain structure changes from dendritic to cellular, achieving the effect of refining the grain size. 

(2) Compared with the HEA coating, the addition of WC produces more hard compounds in the coating, which increases the microhardness of the coating from 470 HV to 560 HV and decreases the friction coefficient from 0.61 to 0.29. The wear amount of the coating decreases significantly, the wear form changes from adhesive wear to more obvious abrasive wear, and the wear resistance of the coating is greatly improved. However, the wear resistance is not stable, and the wear resistance of the coating decreases sharply when the hard compound is worn. After the addition of 2% CeO_2_, although the average friction coefficient increases to 0.46, the friction coefficient is relatively stable in the whole wear experiment, and the wear resistance of the coating is reliable and has best shape after wear. The wear features are only slight grooves, with no spalling, cracks, and other defects, prolonging the service life of the workpiece.

(3) Compared to the HEA/WC coating, the addition of 2% CeO_2_ makes the charge transfer more difficult at the interface between the electrolyte and the electrode, and the capacitive arc diameter increases significantly. Moreover, the HEA/WC + CeO_2_ coating passivates at a voltage of −0.684 V, the passivation zone of the polarization curve of the coating is more obvious, and the corrosion current density is the lowest. The phase angle is increased from 51° to 60°, and the polarization impedance is increased by 360 Ω · cm^2^, resulting in a denser passivation film during corrosion, a relatively lower corrosion rate, and better corrosion resistance.

## 5. Future Work

We will continue to explore various process methods and various powder additives for the FeCoNiCrMo0.2 coating to explore more excellent and stable performance and to prepare coating materials with more comprehensive performance, which will in turn improve the surface wear and corrosion resistance required for 316L parts in actual working conditions. In the future, we can also consider the use of such comprehensive performance powder materials in metal 3D printing to produce metal parts with high strength, high wear resistance, and high corrosion resistance.

## Figures and Tables

**Figure 1 nanomaterials-13-01104-f001:**
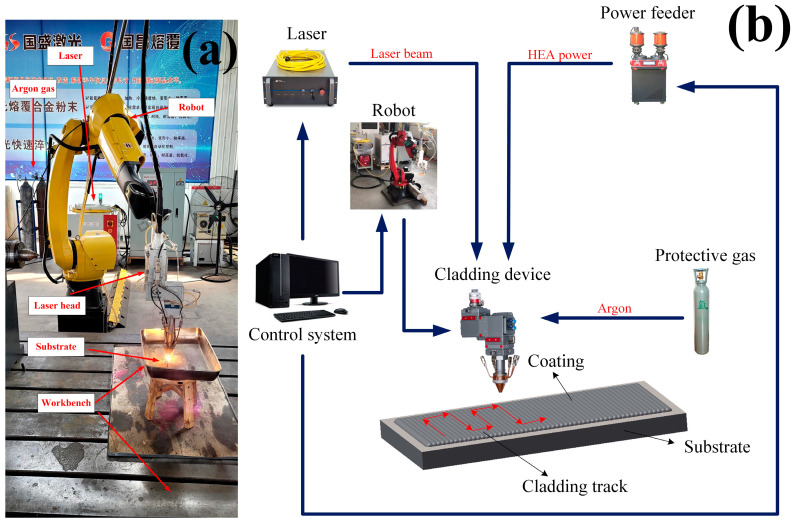
Laser cladding systems and equipment: (**a**) laser cladding equipment; (**b**) component modules of the fusion cladding system.

**Figure 2 nanomaterials-13-01104-f002:**
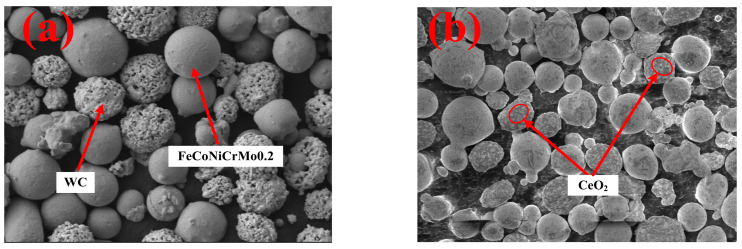
The morphology of the two composite powders: (**a**) FeCoNiCrMo0.2 + WC powder; (**b**) FeCoNiCrMo0.2 + WC + CeO_2_ powder.

**Figure 3 nanomaterials-13-01104-f003:**
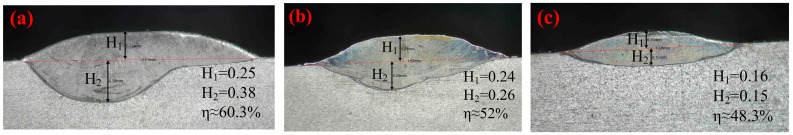
Cross-sectional morphology of the three coatings: (**a**) FeCoNiCrMo0.2 coating; (**b**) FeCoNiCrMo0.2 + 32%WC coating; (**c**) FeCoNiCrMo0.2 + 32%WC + 2%CeO_2_ coating.

**Figure 4 nanomaterials-13-01104-f004:**
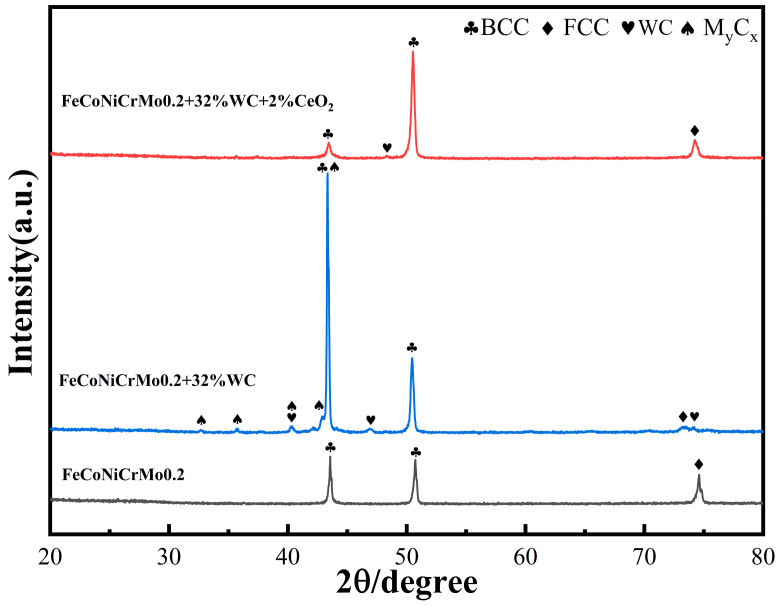
XRD pattern of three coatings.

**Figure 5 nanomaterials-13-01104-f005:**
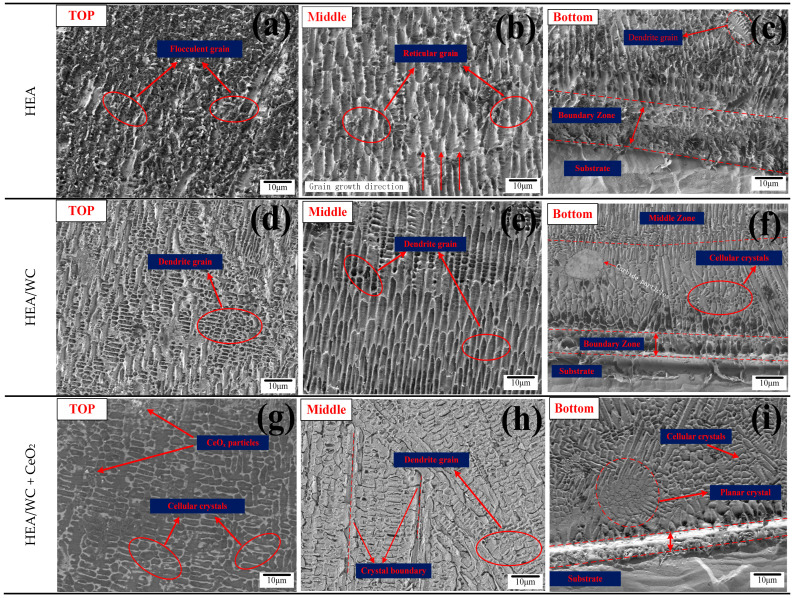
The microscopic morphology of the three coatings: (**a**–**c**) HEA; (**d**–**f**) HEA/WC; (**g**–**i**) HEA/WC + CeO_2_.

**Figure 6 nanomaterials-13-01104-f006:**
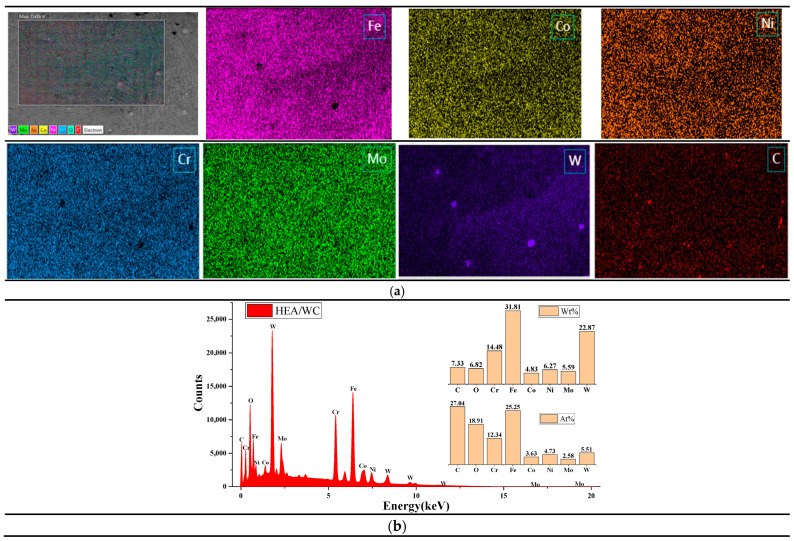
(**a**). The energy spectrum analysis results of each element of the HEA/WC coating. (**b**). The distribution of each element of the HEA/WC coating.

**Figure 7 nanomaterials-13-01104-f007:**
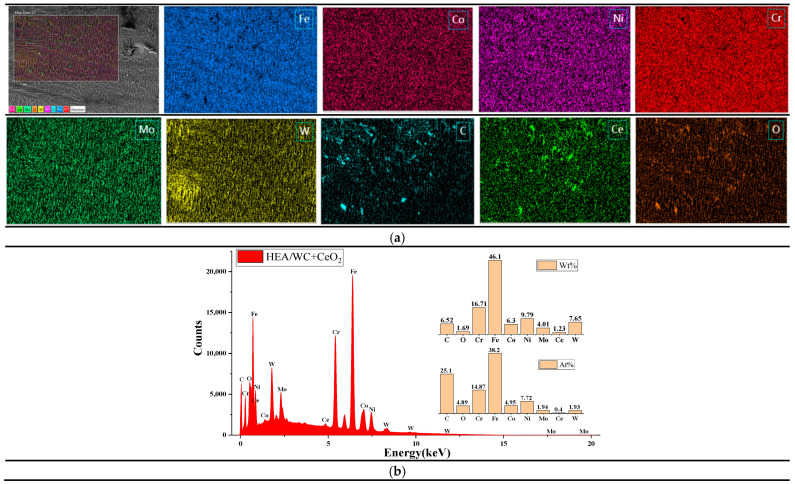
(**a**). The energy spectrum analysis results of each element of the HEA/WC + CeO_2_ coating. (**b**). The distribution of each element of the HEA/WC + CeO_2_ coating.

**Figure 8 nanomaterials-13-01104-f008:**
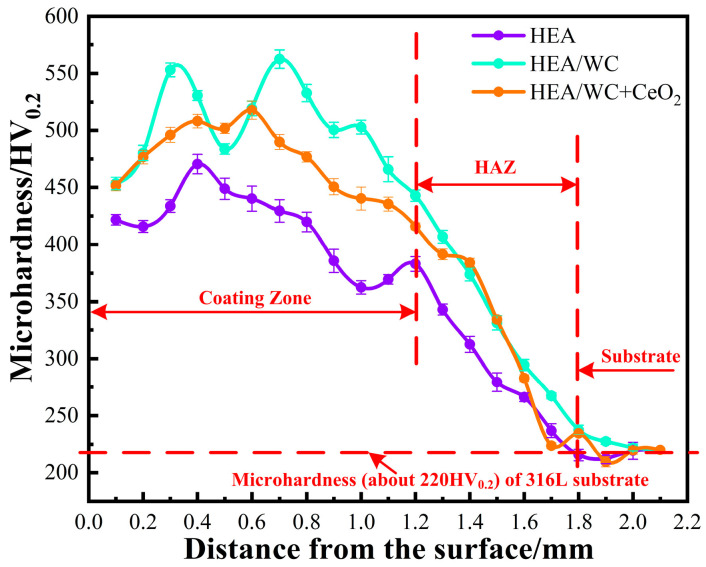
Microhardness distribution curves of different coatings.

**Figure 9 nanomaterials-13-01104-f009:**
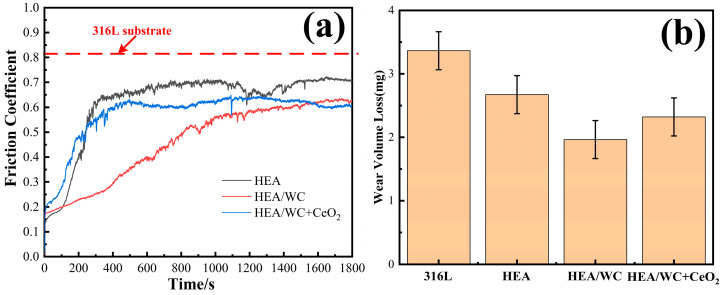
The results of the wear test: (**a**) friction coefficient curves; (**b**) wear amount.

**Figure 10 nanomaterials-13-01104-f010:**
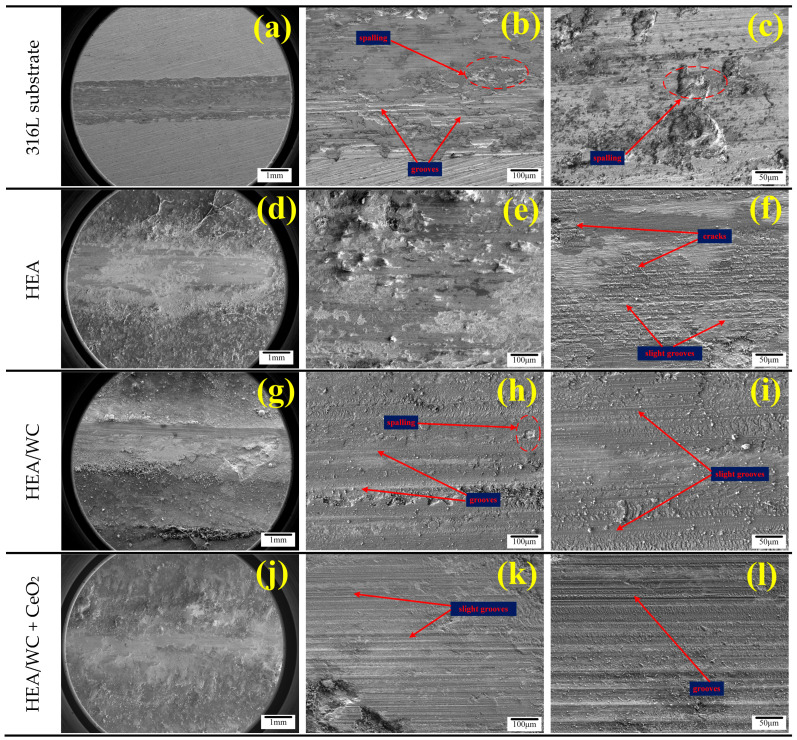
Microscopic morphology of the substrate and the three coating surfaces after wear: (**a**–**c**) 316L substrate; (**d**–**f**) HEA; (**g**–**i**) HEA/WC; (**j**–**l**) HEA/WC + CeO_2_.

**Figure 11 nanomaterials-13-01104-f011:**
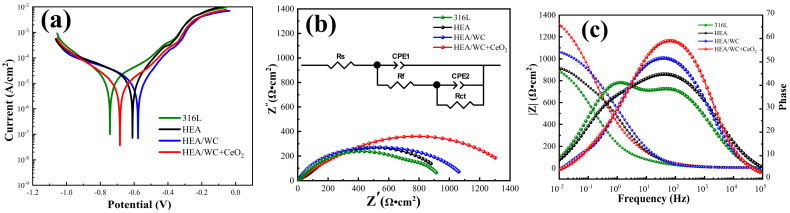
Curves of electrochemical corrosion results: (**a**) polarization curves; (**b**) Nyquist diagram; (**c**) Bode plot.

**Table 1 nanomaterials-13-01104-t001:** Chemical composition of FeCoNiCrMo0.2 powder (wt.%).

Fe	Mo	Ni	Co	Cr
23.02	7.75	23.92	20.99	Bal.

**Table 2 nanomaterials-13-01104-t002:** Technical parameters of cladding process.

Power (W)	Spot Diameter (mm)	Scanning Speed (mm·min^−1^)	Powder Delivery Capacity (g·min^−1^)	Defocusing Amount (mm)	Overlap Rate
2800	2.0	1000	15	16	60%

**Table 3 nanomaterials-13-01104-t003:** Chemical composition of CeO_2_ composite powder with different contents.

Number	Powder Components
1	FeCoNiCrMo0.2 + 32%WC
2	FeCoNiCrMo0.2 + 32%WC + 2%CeO_2_
3	FeCoNiCrMo0.2 + 32%WC + 4%CeO_2_
4	FeCoNiCrMo0.2 + 32%WC + 6%CeO_2_
5	FeCoNiCrMo0.2 + 32%WC + 8%CeO_2_

**Table 4 nanomaterials-13-01104-t004:** Enthalpy of mixing between elements (KJ · mol^−1^).

Fe	Co	Ni	Cr	Mo	W	C
Fe	−1	−2	−1	−2	0	−50
	Co	0	−4	−5	−1	−42
		Ni	−7	−7	−3	−39
			Cr	0	1	−61
				Mo	0	−67
					W	−60
						C

**Table 5 nanomaterials-13-01104-t005:** Experimental results of self-corrosion performance in pH = 7.

Sample	E_corr_/V	I_corr_/A · cm^−2^
316L substrate	−0.744	5.31 × 10^−7^
HEA	−0.610	7.27 × 10^−8^
HEA/WC	−0.577	7.04 × 10^−8^
HEA/WC + CeO_2_	−0.684	3.68 × 10^−8^

**Table 6 nanomaterials-13-01104-t006:** The fitting results of components in the equivalent circuit.

Sample	R_s_/(Ω · cm^2^)	C_CPE1_/(Ω^−1^ · cm^−2^ · s^n^)	n_1_	R_f_/(Ω · cm^2^)	C_CPE2_/(Ω^−1^ · cm^−2^ · s^n^)	n_2_	R_ct_/(Ω · cm^2^)	χ^2^	R_p_/(Ω · cm^2^)
316L substrate	6.313	8.454 × 10^−4^	0.584	189.8	6.102 × 10^−4^	0.769	811.2	3.647 × 10^−3^	1.007 × 10^3^
HEA	6.14	3.708 × 10^−4^	0.665	819.2	2.356 × 10^−3^	0.932	225.7	3.137 × 10^−3^	1.051 × 10^3^
HEA/WC	4.383	2.063 × 10^−4^	0.751	628.1	1.549 × 10^−3^	0.593	487.8	2.191 × 10^−3^	1.120 × 10^3^
HEA/WC + CeO_2_	6.401	5.353 × 10^−4^	0.585	723.3	7.288 × 10^−4^	0.687	750.5	2.501 × 10^−3^	1.480 × 10^3^

## Data Availability

Data will be made available upon request.

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
