# Peer review of "Effects of Nano-CeO2 on Microstructure and Properties of WC/FeCoNiCrMo0.2 Composite High Entropy Alloy Coatings by Laser Cladding"

_nanomaterials, 2023, doi:10.3390/nano13061104_

Round 1
Reviewer 1 Report
The manuscript entitled “nanomaterials-2291261” dealing with laser cladding has been reviewed. The paper has been nicely written but needs significant improvement. Please follow my comments.
1. What is the main research question for this research work?
2. Page 2: The size of the font is too small.
3. What is the future direction of this work?
4. How did you select the parameters in Table 3?
5. How the cross-section in Fig 3 was produced. More detail is required.
6. Laser absorptivity is important which shows the quality of the parts and transition from keyhole to conduction mode. Please read and add the following ref in this area. “The effect of absorption ratio on meltpool features in laser-based powder bed fusion of IN718”.
7. Laser has many advantages over the conventional manufacturing method which can be highlighted in your paper. Please read the following manuscript and add it to the literature to show how the laser is comparable with conventional manufacturing. “Laser subtractive and laser powder bed fusion of metals: review of process and production features”.
Reviewer 2 Report
Reviewer remarks to the article:
Effects of nano-CeO2 on microstructure and properties of WC/FeCoNiCrMo0.2 composite high entropy alloy coatings by laser cladding
This paper is very well constructed. The results of investigations are valuable and very interesting from the point of view of materials with nanoparticles properties used for coatings applied to steel. This is the correct planned and done very valuable scientific work. The authors applied a methodical apparatus adequate to the assumed goals. But the following comments should be addressed before considering of publication:
1) It would be good at the end of chapter 1 to write something synthetic about the research gap as a justification for the undertaken experimental research.
2) The descriptions in some photos are illegible.
3) The descriptions in Fig 11 are unreadable.
4) There are errors in the references, for example in [2].
Author Response
Professor, I have replied to all your comments in the attached file. Please see the attachment.
